# High-throughput chiral copper foils by curved-surface confinement recrystallization

Deping Huang[1,2,8], Zhancheng Li[1,2,8], Yinwu Duan[1,3], Xin li[1,2], Yongna Zhang[1,2], Jiaxing Dong[4], Guilin Wu [5], Xiaoxu Huang [5], Leining Zhang[6] ✉, Feng Ding [7] ✉ & Haofei Shi [1,2] ✉

Chiral metal surfaces play a pivotal role in enantioselective catalysis, sensing, and spintronics, yet their scalable fabrication remains challenging due to a reliance on chiral templates or molecular precursors, which limits both throughput and precise control of crystallographic orientation. Here, we report a high-throughput method for fabricating chiral copper surfaces via curved-surface confinement recrystallization. This approach exploits curvature-driven abnormal grain growth to transform polycrystalline foils into large-area crystals with continuously graded high-index surfaces. Systematic control of the curvature during annealing enabled the creation of a library of chiral copper surfaces, providing high-throughput and surface templates with defined chirality. Through manipulation of the initial crystal orientation and curvature, single crystals with tailored surface orientations can be reached. The intrinsic chirality of these surfaces is confirmed by circular dichroism spectroscopy and model asymmetric reactions. Furthermore, we demonstrate the transfer of chirality to epitaxial two-dimensional materials, exemplified by the growth of chiral graphene. This work provides a scalable platform for producing designer chiral surfaces, enabling future advances in asymmetric catalysis and chiral device engineering.

Chiral metal surfaces play a crucial role in enantioselective processes, making them highly valuable for applications in asymmetric catalysis[1–4], chiral sensing[5–7], and chiral spintronics[8–11]. Metals generally exhibit chiral surfaces unless their surfaces align with the mirror planes of the crystal[12–16]. Specifically, chiral surfaces are observed in high Miller index $(h\ k\ l)$ orientations where $h \times k \times l \neq 0$ and $h \neq k \neq l$[17,18]. The ability to prepare large-area, configurationally stable, and reproducible chiral metallic surfaces is of great importance for the advancement of enantioselective applications. Current fabrication strategies, such as

heteroepitaxial growth[19–22], chiral imprinting[23–25], and nanoparticle synthesis[26–29], remain inherently constrained by their reliance on chiral precursors or templates. This dependency restricts scalability in surface orientation control and high-throughput fabrication.

Abnormal grain growth during annealing process of polycrystalline metal foils has been validated as an effective method for fabricating large-area single-crystal metal foils. Beyond low-index facets such as Cu(111)[30–32], high-index crystal facets can be achieved through thermal strain moderation[33], edge incision introduction[34], or seeded

[1]Chongqing Institute of Green and Intelligent Technology, Chinese Academy of Sciences, Chongqing, PR China. [2]University of Chinese Academy of Sciences (UCAS), Beijing, PR China. [3]National Engineering Research Center for Instrument Functional Materials, Chongqing, PR China. [4]School of Chemistry and Chemical Engineering, Southwest University, Beibei, Chongqing, PR China. [5]International Joint Laboratory for Light Alloys (MOE), College of Materials Science and Engineering, Chongqing University, Chongqing, PR China. [6]Beijing Key Laboratory of Intelligent Molecular Materials and High-throughput Manufacturing, School of Chemistry and Chemical Engineering, Beijing Institute of Technology, Beijing, PR China. [7]Research Division of Advanced Materials, Suzhou Laboratory, Suzhou, PR China. [8]These authors contributed equally: Deping Huang, Zhancheng Li. ✉e-mail: leiningzhang@bit.edu.cn; dingf@szlab.ac.cn; shi@cigit.ac.cn

growth[35]. However, precise control over arbitrary chiral surfaces remains a challenge. Current approaches, such as heteroepitaxy, nanoparticle synthesis, and annealing process, are inherently limited by low throughput and orientation randomness. Such constraints hinder their industrial adoption, necessitating alternative fabrication strategies.

Herein, we propose a confinement recrystallization method for fabrication of high-throughput chiral copper surfaces via curved-surface confinement. This approach enables programmable crystal orientation control during annealing, where curved-surface confinement drives non-equilibrium abnormal grain growth, leading to the formation of a large-area single-crystal copper foil with a graded surface. The diverse chirality of the graded surface was confirmed by circular dichroism response and asymmetric catalytic activity. Serving as a substrate, direct chirality transfer from copper foil to two-dimensional (2D) materials (e.g., graphene) is demonstrated. In addition, by engineering curvature-flat surface architectures, tailored single-crystal copper foils with a designed orientation were achieved. This technique provides an efficient route for large-scale production of chiral surfaces and single crystals with desired orientations.

## Results

### Chiral surface gradients in grain-boundary-free copper foils

Figure 1a schematically illustrates our proposed confinement annealing recrystallization strategy. During curved-surface confinement annealing, abnormally grown grains develop orientation gradients through a formation energy minimization mechanism, enabling the synthesis of grain-boundary-free copper foils with surface orientation gradients. In contrast, during flat-surface annealing (Fig. 1b), abnormal grains expand laterally, ultimately forming large-area single crystals. Polycrystalline copper foils were placed along the inner side of the quartz tube for curved-surface confinement recrystallization annealing (Fig. 1c). To enhance the optical contrast of the resulting graded chiral surface, the curved copper foil was flattened and heated on a hotplate at 200 °C in air for 30–60 s to oxidize the surface. The oxidized surface in Fig. 1d displays continuously varying color contrasts, implying the variation of the surface orientations of the copper foil.

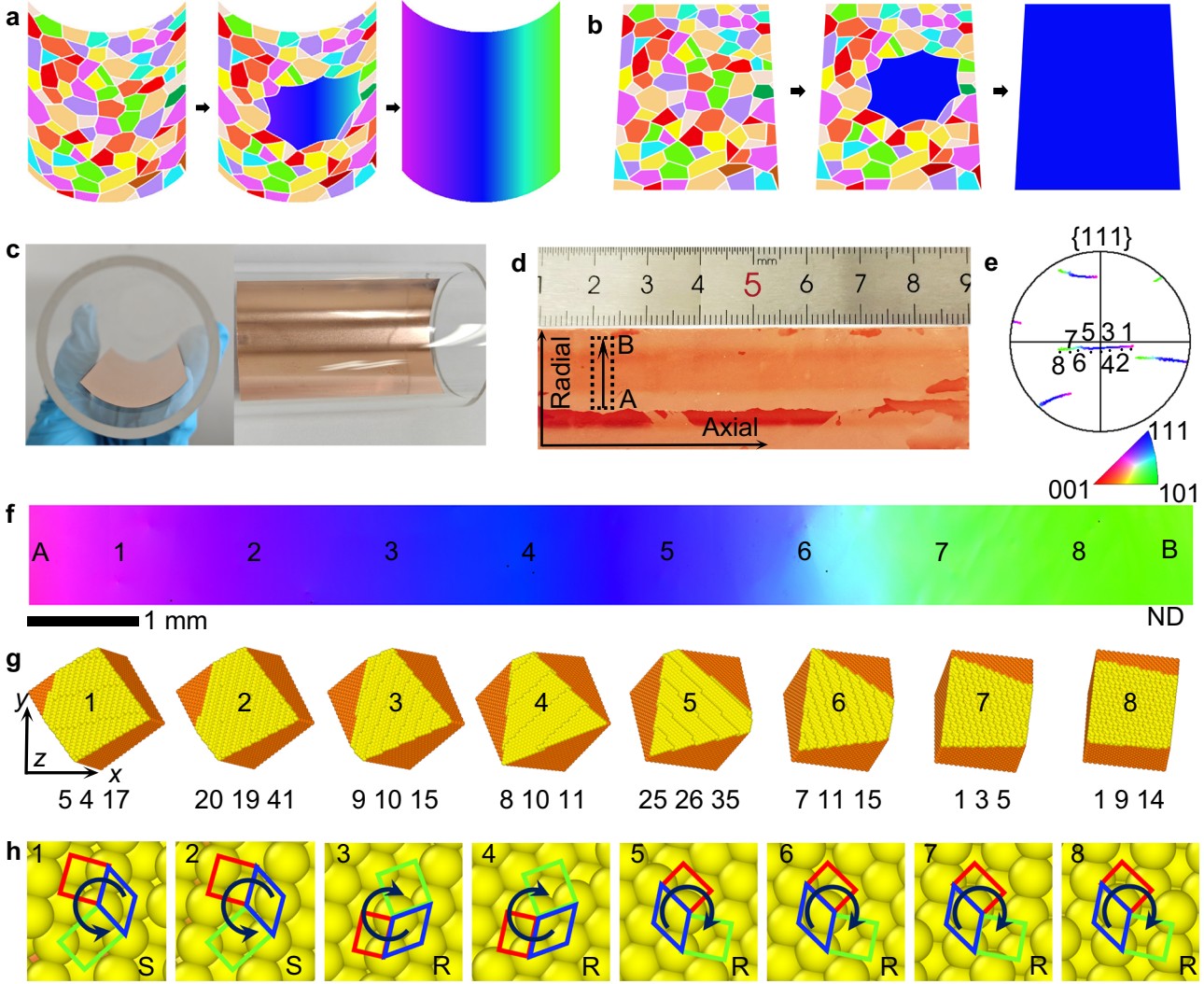

**Fig. 1 | Characterization of copper foil produced by confinement recrystallization.** Schematic comparison of surface orientation evolution in recrystallized copper foils under two annealing conditions: **a** Curved-surface. **b** Flat-surface. **c** Photograph of copper foil confinement annealing in quartz tube. **d** Optical image of the annealed copper foil within a 24 mm diameter quartz tube. **e** EBSD pole figure of the copper foil for the {111} plane from positions A-B (marked in **d**). **f** EBSD IPF map in normal direction of the copper foil from positions A-B. The scale in panel **e** applies to (**a–f**). **g, h** The atomic arrangements and chirality of the surfaces corresponding to position 1-8, indexed as Cu(5 4 17)$^S$, Cu(20 19 41)$^S$, Cu(9 10 15)$^R$, Cu(8 10 11)$^R$, Cu(25 26 35)$^R$, Cu(7 11 15)$^R$, Cu(1 3 5)$^R$, and Cu(1 9 14)$^R$. The yellow color represents the copper atoms formed atomic steps on the surface. The colors red, green, and blue square in 1 **h** represent the (001), (110), and (111) crystallographic planes, respectively.

Electron backscatter diffraction (EBSD) was employed to analyze the surface orientation of obtained copper foil. We define the direction along the tube diameter as the axial direction, and the direction along the curvature as the radial direction. The pole figures (Fig. 1e), orientation distribution function plots (Supplementary Fig. 1a), and inverse pole figure (IPF, Supplementary Fig. 1b) derived from the EBSD measurements all reveal that the surface orientation varies continuously from position A to position B along the radial direction within the measured region. The IPF map of copper foil in normal direction further demonstrates that a smooth color transition across a distance of approximately 10 mm, without the formation of grain boundaries (Fig. 1f), verifying the uniform copper foils with orientation gradients. High-resolution transmission electron microscope (HRTEM) characterization also confirms the preservation of uniform lattice structures on the gradient copper foils (Supplementary Fig. 1c). The structural evolution of copper foil was driven by the minimization of surface energies, grain boundary energies, and elastic energies. During our proposed confinement annealing process, the copper foil adopted to the arc shape of the tube, storing elastic energy that was later released. For the fully-relaxed structure, two stable configurations were possible: (i) a polycrystal with grain boundaries accommodating the curvature, or (ii) a single crystal with a gradually rotating surface normal. Despite the graded surface of single-crystal copper foil contains high-energy surface regions, its overall formation energy remains lower than that of polycrystals. Consequently, abnormal grain growth was promoted within the single-crystal region, leading to the formation of a large-area single-crystal copper foil with a graded surface (Supplementary Fig. 2).

Curved-surface confinement is the critical factor in the formation of orientation gradients during the abnormal grain growth. Analysis of surface orientation variations (Supplementary Fig. 3a) demonstrates that the surface orientations remain invariant along the axial direction (Supplementary Fig. 3b), whereas a gradual transition in surface orientations is observed along the radial direction (Supplementary Fig. 3c). Meanwhile, copper foils obtained via flat-surface annealing exhibit a uniform orientation with misorientation less than 2° (Supplementary Fig. 4). These results indicate that curved-surface confinement drives self-adaptive transformations of surface orientations along the geometrically constrained curved surfaces during abnormal grain growth. To investigate the orientation change occurs in only surface or the whole foil, we thinned the annealed copper foil via electrochemical polishing. Surface orientation remained consistent across all thicknesses (Supplementary Fig. 5), confirming bulk abnormal grain growth throughout the foil.

The gradient substrate exhibits continuous chirality variations. Surface models constructed based on Euler angle data reveal distinct atomic chiral patterns at eight equally spaced positions across the measured region (Fig. 1g and Supplementary Fig. 1d–f). A transition from sinister (*S*) to rectus (*R*) chirality is observed at these positions (Fig. 1h), demonstrating that curved-surface confinement also influences the chiral characteristics of copper foil surfaces. Atomic force microscopy (AFM) characterizations directly compared the surface morphology before and after curved annealing. The as-received copper foil exhibits rolling marks and a smooth, featureless surface. After annealing, well-defined step-terrace structures emerge across the surface (Supplementary Fig. 6). Scanning tunneling microscopy (STM) images further resolve kinked step edges (Supplementary Fig. 7), a structural signature indicative of surface reconstruction and consistent with chiral morphology.

### Curvature-engineered orientation and chirality gradients
To explore the effect of curvature on the surface orientations and chirality, we annealed copper foils under different curvatures ($\kappa = 0.011$–$0.143$ mm$^{-1}$) with diameters of quartz tubes from 187 to 14 mm (Fig. 2 and Supplementary Fig. 8). The misorientation of surface orientations between two positions along the radial direction is defined as the graded angle, and the central angle between two positions is defined as the radian (Figure 2a). Figure 2b displays EBSD IPF maps of copper foils with identical arc lengths annealed under varying curvatures, revealing distinct surface orientation gradients. As curvature decreases, the color gradient diminishes, suggesting a slower rate of surface orientation evolution. A linear relationship was fitted between the graded angle and radian on copper foils annealed under various curvatures (Fig. 2c and Supplementary Fig. 9a, b). Within the same arc length, the radian decreases with the decreasing of curvatures (Fig. 2d and Supplementary Fig. 9c). When curvature approaches zero, the obtained copper foil exhibits single-crystal characteristics (Supplementary Fig. 4). To further verify the reproducibility of the method, multiple independent experiments were performed under a fixed curvature condition ($\kappa = 0.029$ mm$^{-1}$). The linear relationship between the graded orientation angle and curvature was consistently reproduced across multiple independent trials (Supplementary Fig. 10), providing additional statistical confirmation of the robustness of our confinement annealing approach. Similar quantitative relationships were observed in copper foils with different thicknesses of 46 μm (Supplementary Fig. 11), demonstrating the universal applicability of the approach.

The dependence of chirality degree on the surface Miller indices was examined. In Fig. 2e, f, the atomic configurations and chirality degrees of surfaces are displayed in the inverse pole figure. The surfaces denoted by the edges of the stereographic triangle in the inverse pole figure, with the Miller indices of (*m m n*), (*m n n*), and (0 *m n*), have a mirror symmetry and therefore are achiral. In contrast, copper surfaces located in the inner part have kinked steps and are truly chiral. The deviation of a chiral surface from its nearest achiral counterpart is defined as the degree of chirality. The greater a chiral surface deviates from its nearest achiral counterpart, the higher its degree of chirality. Figure 2g reveals that the two sides of the graded surface across a mirror-symmetry plane exhibit opposite chirality. It is worth noting that a graded copper foil with a desired degree of chirality can be achieved by strategically designing the magnitude of gradient (out-of-plane orientation) and the direction of gradient (in-plane orientation) in copper surface.

The chirality of the curved-annealed copper surfaces was assessed using circular dichroism (CD) spectroscopy. Measurements (Fig. 3a–c) reveal distinct CD signals corresponding to different surface configurations. For instance, the achiral Cu(0 27 96) surface shows a near-zero response, whereas chiral surfaces such as Cu(60 32 73)$^S$ and Cu(14 67 73)$^R$ exhibit pronounced signals of opposite chirality (ca. 20 mdeg and - 30 mdeg, respectively). The CD response for each surface aligns well with its chirality.

### Chiral copper: replication, epitaxy and asymmetric catalysis
The obtained Cu foil with a gradient orientation provides a seed library for the fabrication of chiral single-crystal Cu foil with a desirable surface index. By extracting small piece from the gradient copper surfaces, replication of large-area chiral copper foils can be achieved. Selected seeds were individually placed on pre-oxidized surfaces of polycrystalline copper foils. During annealing treatment, the seeds induced abnormal grain growth in the underlying substrate, ultimately forming large-area chiral foils. The EBSD IPF maps and corresponding IPFs of regions A, B, and C (Fig. 3d–g) confirm replication of the seed surface orientations in the obtained single crystals.

The chiral substrates hold potential for diverse applications, leveraging the enantioselective properties inherent to chiral surfaces. For example, using the obtained gradient copper foils as catalytic surfaces, we have verified the influence of chiral surfaces on graphene epitaxy. The photograph and corresponding EBSD IPF map of the gradient substrate used for graphene epitaxy are shown in Supplementary Fig. 12a–c. The atomic structure of the chiral surface in Fig. 4a

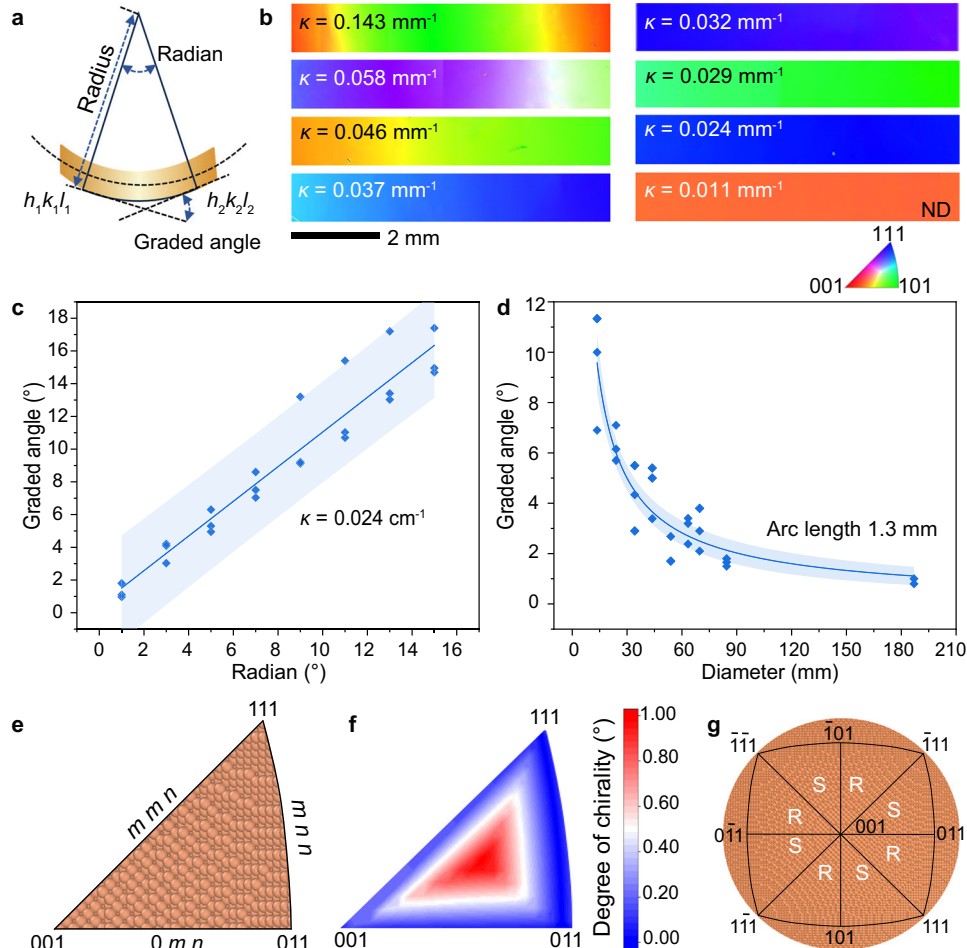

**Fig. 2 | The effect of curvature on surface orientations evolution. a** Schematic illustration of the "graded angle" and the "radian". **b** EBSD IPF maps of copper foils (arc length: 1.3 mm) annealed under different curvatures (κ). ND denotes the sample normal direction. **c** Relationship between the graded angle and radian for a copper foil with a curvature of 0.02 mm⁻¹. Individual data points represent results from three independent annealing experiments ($n = 3$), compared to flat-surface annealing controls. The relationship is well described by a significant linear regression (slope = 1.06 ± 0.06; adjusted $R^2$ = 0.92; ANOVA, $p = 4.42 \times 10^{-6}$). The solid line represents the best-fit line, and the shaded band indicates its 95% confidence interval. **d** Relationship between the graded angle and quartz tube diameter. Individual data points represent results from three independent annealing

experiments ($n = 3$), compared to flat-surface annealing controls. The data were well fitted by an allometric model (adjusted $R^2$ = 0.85; ANOVA, $p = 2.41 \times 10^{-17}$). The solid line represents the best-fit curve, with the shaded band indicating the 95% confidence interval. **e** The atomic configurations displayed in the inverse pole figure. The surfaces denoted by the edges of the stereographic triangle have a mirror symmetry and therefore are achiral. **f** The degree of chirality of Cu surfaces, which is defined as the deviation of a chiral surface from its nearest achiral counterpart. **g** Chirality of the graded copper surfaces denoted in inverse pole figure. Orange spheres represent copper atoms. Source data are provided in Source Data file.

exhibit chiral characteristics, with the surface orientations gradually evolving from Cu(8 7 32)$^R$ to Cu(7 8 32)$^S$. Statistical results from optical images of graphene grains reveal a strong correlation between their morphological evolution, directions, and the chiral characteristics of the epitaxial substrate surface (Supplementary Fig. 12d). From Cu(8 7 32)$^S$ to Cu(7 8 32)$^R$, graphene grains shift from grain-A dominance to grain-B dominance (Fig. 4b, c). Notably, grains A and B exhibit distinct chirality (inset of Fig. 4c).

Angle-resolved polarized Raman spectroscopy on the graphene grains reveals that the graphene grain edges comprise a hybrid of zigzag and armchair configurations. The SEM images and Raman mapping of the graphene grain without polarization are shown in Fig. 4d, e and Supplementary Fig. 13a. Under linearly polarized light (Fig. 4f and Supplementary Fig. 13b, c), the polarization-dependent from D-band intensity showed a polarization anisotropy ratio $\rho$ ($I(D)_{min}/I(D)_{max}$) around 0.58 ± 0.13 under 0°-90° polarized light, indicating a chirality edge structure composited by zigzag and armchair[36–39]. Furthermore, a distinct intensity variation was observed under left-handed (LCP) and right-handed circular polarization (RCP)

as shown in Fig. 4g and Supplementary Fig. 13d, e, providing complementary evidence for the chiral and hybrid nature of the graphene edges.

The graphene epitaxy experiment validates the spatial correlation between the chiral gradient of the Cu foil surface and the chiral structure of graphene. Epitaxial growth establishes a curvature-orientation-chirality correlation model for predictive design of 2D chiral materials. Validation confirms that the Cu($h\,k\,l$) $R \rightarrow S$ chiral-gradient substrate enables accelerated high-throughput screening of optimal chiral configurations, shortening chiral-surface-based research cycles.

The asymmetric catalysis experiments demonstrated that curved-surface annealed copper foils can effectively catalyze the oxidation of racemic 1-phenylethanol while inducing enantioselectivity. Gas chromatography-mass spectrometry (GC-MS) spectrum (Supplementary Figs. 14 and 15) and chiral high-performance liquid chromatography (HPLC) in Supplementary Fig. 16 confirmed an enantiomeric enrichment (enantiomeric excess, ee = 5.82%) was achieved when the intrinsically chiral Cu(06 69 72) foil was employed as a catalyst. This value is significantly higher than that of the starting material and the

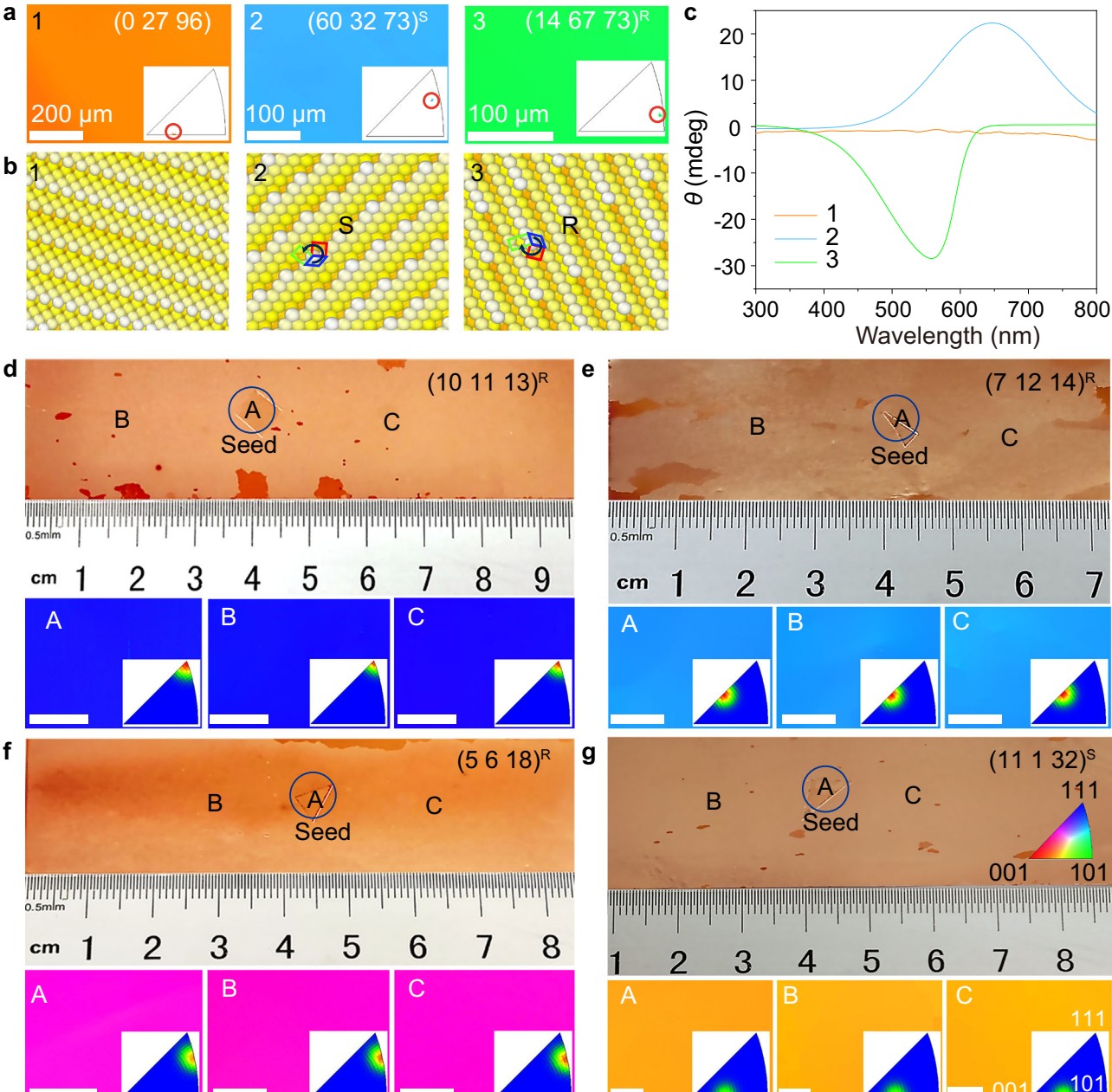

**Fig. 3 | Characterization of chirality of the copper surfaces and seeded growth of chiral copper foils. a** EBSD IPF maps of the copper surfaces prepared for CD measurements. The different colors represent the respective positions in the color scale corresponding to single-crystal Cu(0 27 96), Cu(60 32 73)$^S$ and Cu(14 67 73)$^R$. The insets are IPF plots for normal direction. **b** Atomic surface models and schematic analysis illustrating the intrinsic chirality of the crystal planes corresponding to those in (**a**). **c** CD spectra measured from the copper foils. Each spectrum is colored according to its corresponding surface orientation in (**a**). **d–g** Representative photographs and corresponding EBSD IPF maps of chiral copper foils fabricated by seeded growth: **d** Cu(10 11 13)$^R$; **e** Cu(7 12 14)$^R$; **f** Cu(5 6 18)$^R$; **g** Cu(11 1 32)$^S$. Scale bars: 200 μm (**d–f**), 50 μm (**g**). The color scale in (**g**) applies to panels (**a**, **d–f**). The insets are IPFs for the normal direction. Source data are provided in the Source Data file.

control experiment using the achiral Cu(5 6 6) foil. This result provides direct experimental evidence that the chirality-imparted copper substrate exhibits asymmetric catalytic activity.

## Discussion

Building upon our established confinement technique for copper foil annealing, this methodology holds promise for generating substantially diversified surface orientations by conical-surface confinement annealing (Supplementary Fig. 17). Driven by a gradient confinement effect arising from continuously varying curvature along the conical axis, this approach enables near-complete coverage of the IPF (Supplementary Fig. 17e). These surface orientations radiate fan-wise from the cone apex, establishing an exhaustive orientation library that spans >90% of the orientations. Crucially, our technique generates a comprehensive seed library covering diverse orientations in a single experiment, thereby laying the foundation for scalable chiral surface preparation.

Fundamentally transcending traditional facet control paradigms that depend on strain energy manipulation, surface energy minimization, or stochastic processes, the curved-surface confinement strategy leverages kinetically controlled growth to eliminate orientation uncertainty. This approach accomplishes a paradigm shift in surface

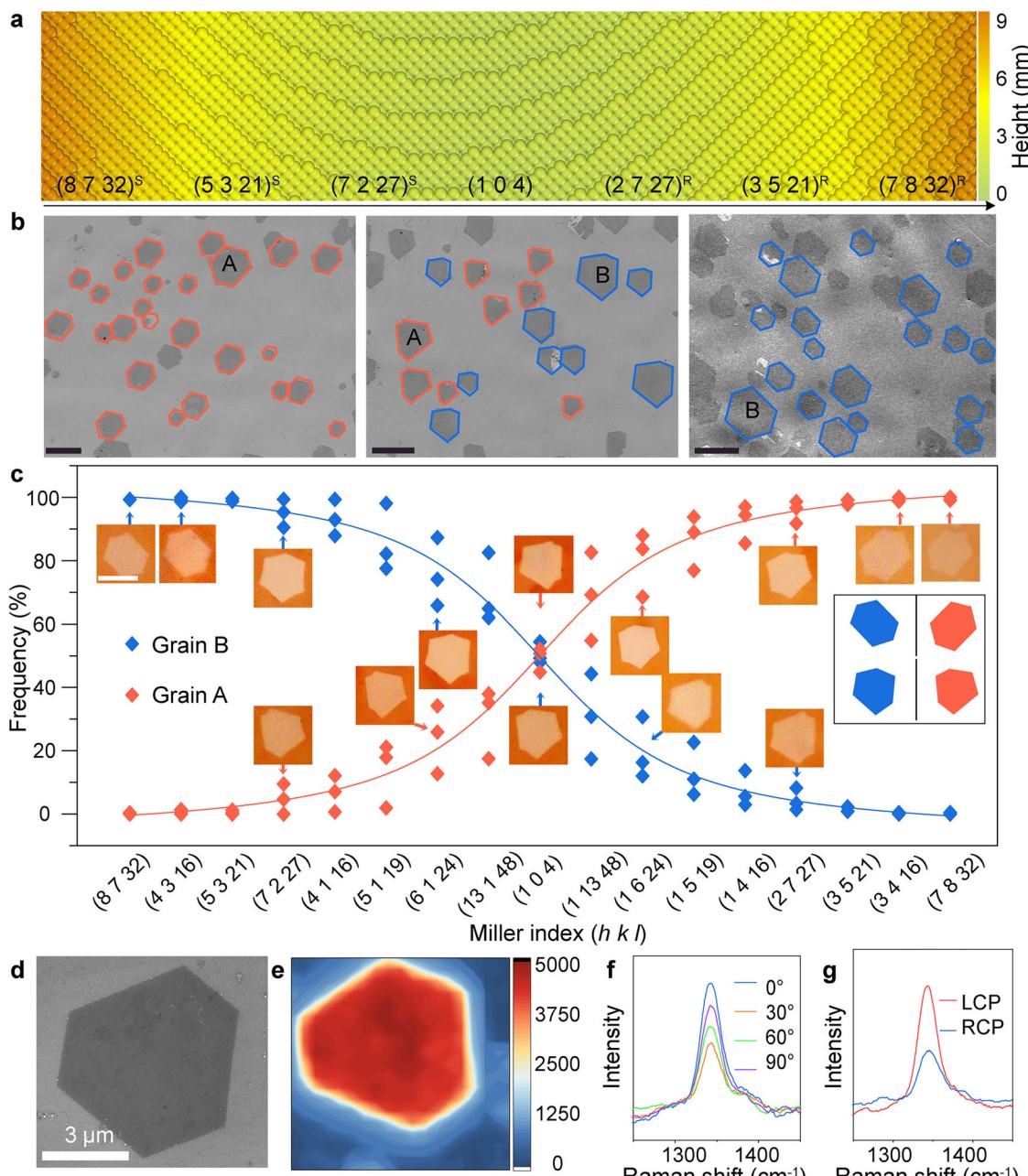

**Fig. 4 | Epitaxial graphene on the obtained gradient copper foil. a** The atomic structure of the chiral surface for graphene epitaxy, with index evolving from Cu(8 7 32)$^S$ to Cu(5 3 21)$^S$, Cu(7 2 27)$^S$, Cu(1 0 4), Cu(2 7 27)$^R$, Cu(3 5 21)$^R$, and Cu(7 8 32)$^R$. **b** The typical SEM images of graphene grains. Scale bar: 10 μm. **c** Statistical distribution of the occurrence frequency for graphene grain types A and B along the gradient substrate. Individual data points were obtained from three independent graphene growth experiments ($n = 3$). The frequency for both grain A and grain B was described by a Langevin function model, with adjusted $R^2$ values > 0.97. The overall fits were highly significant (ANOVA, $p = 2.88 \times 10^{-39}$ for grain A; $p = 1.58 \times 10^{-39}$ for grain B). Scale bar for inset images: 10 μm. **d** SEM images of the graphene grain for angle-resolved Raman testing. **e** Raman 2D-band mapping of the graphene grain. The color intensity corresponds to the relative Raman scattering intensity (in arbitrary units, arb. u.). Corresponding D-band intensity comparison under 0°–90° polarized light (**f**) and under LCP and RCP light (**g**). Source data are provided in Source data file.

orientation engineering from localized exploration to global mapping. When integrated with roll-to-roll processing, this methodology unlocks the potential for industrial-scale production of chiral metal foils, with transformative applications in catalysis membranes, low-impedance electronics, and epitaxial growth substrates.

Through modified curved-surface confinement annealing, this methodology can be further extended to achieve customized control over surface orientations in large-area single-crystal copper foils. By applying graded curved-to-flat confinement and selecting initial grain orientations, we demonstrate programmable crystal orientation control in planar regions. This enables continuous transitions between arbitrary surface orientations, as shown for Cu(6 1 7) to Cu(2 3 16) evolution (Supplementary Fig. 18). Our technique consequently advances surface orientation engineering from trial-and-error methods to an on-demand programmable design paradigm.

In summary, we present a curved-surface confinement recrystallization technique that enables the fabrication of grain-boundary-free copper foils with chiral surface orientations and allows for high-throughput crystal orientation control. During annealing, the curved-surface confinement drives non-equilibrium grain evolution, forming

continuous orientation gradients. By systematically varying the confinement curvature, we have constructed a chiral surface orientation library encompassing nearly all surface orientations within a single annealing process. Through precise manipulation of curved-surface confinement, we demonstrate programmable surface orientation control. Graphene epitaxy reveals direct chirality transfer from substrate to 2D materials, confirming the substrate chirality's decisive role. Our method overcomes conventional efficiency and precision limitations in chiral surface preparation and provides a versatile platform for 2D materials and catalysis research.

## Methods

### Materials and reagents

Copper foils were used as substrates: 25 µm thick foil (product #046365) was purchased from Alfa Aesar (Thermo Fisher Scientific, USA), and 46 µm thick foil (99.95% purity) was obtained from Chinalco Shanghai Copper Co., Ltd. (Shanghai, China). All chemical reagents were used as received: phosphoric acid ($H_3PO_4$, 85% in water, analytical grade, Greagent), ethanol ($C_2H_5OH$, 99.9%, Adamas), n-hexane ($C_6H_{14}$, 97.0%, Adamas), ethyl acetate ($C_4H_8O_2$, 99.8%, Adamas), isopropanol ($C_3H_8O$, 99.5%, Adamas), toluene ($C_7H_8$, 99.8%, Adamas), 1-phenylethanol ($C_8H_{10}O$, 98%, Adamas).

### Confinement annealing of copper foil

Polycrystalline copper foil (10 cm × 7 cm) was electrochemically polished using cyclic voltammetry (CHI 1100 C electrochemical workstation, China) at 2.0 V. The polishing was performed in phosphoric acid solution for 10 min, followed by sequential 5 min ultrasonic cleaning in deionized water and ethanol, and was finally dried with $N_2$ gas. The polished foil was placed in confinement fixture (quartz tubes with 14–187 mm diameters and quartz cones) for annealing in a tube furnace. The chamber was evacuated below 1 Pa, purged with argon to atmospheric pressure, and then heated to 1060 °C at 15 °C min⁻¹ under a mixed gas flow of 10 cm³ standard temperature and pressure (STP) min⁻¹ $H_2$ and 200 cm³ STP min⁻¹ Ar. At 1060 °C, the system was held under a gas flow of 100 cm³ STP min⁻¹ $H_2$ and 500 cm³ STP min⁻¹ Ar for 30 min. After ceasing the $H_2$ flow, the furnace was opened, and the sample was then cooling down to room temperature within 30 min (≈35 °C min⁻¹) under 500 cm³ STP min⁻¹ Ar. The resulting curved foil was flattened at room temperature for later characterization.

### Seeded growth of single-crystal copper foil

Seeded growth was performed by cutting a ≈5 mm seed from confinement-annealed copper foil and positioning it on pre-oxidized copper foil. The assembly underwent thermal annealing at 1060 °C under controlled gas flow ($H_2$: 10–50 cm³ STP min⁻¹; Ar: 200 cm³ STP min⁻¹) for 30–60 min, followed by cooling to ambient temperature. Morphological and structural characterization was subsequently conducted.

### Graphene growth on gradient copper foil

After confinement annealing of the copper foil, the temperature was maintained at 1060 °C for direct graphene growth. The graphene growth process used a gas mixture of $H_2$ (100 cm³ STP min⁻¹), Ar (200 cm³ STP min⁻¹), and 2% $CH_4/H_2$ (5-50 cm³ STP min⁻¹) for 10-15 min. Following growth, the furnace cooled to room temperature under continuous Ar flow (200 cm³ STP min⁻¹).

### Asymmetric catalysis experiments

A solution of 50 mg of racemic 1-phenylethanol in 2 mL of toluene was reacted in a hydrothermal autoclave at 210 °C for 8 h using a 1 cm² chiral copper foil as the catalyst. A control experiment was performed under identical conditions with an achiral copper foil of the same size. After the reaction vessel cooled to room temperature, the mixture was collected. The reactants and products were separated by column

chromatography (n-hexane/ethyl acetate, 8:1). The recovered, unreacted 1-phenylethanol was then analyzed by chiral HPLC (OJ-H column; n-hexane/isopropanol, 90:10; flow rate: 1 mL min⁻¹) to determine the enantiomeric excess (ee) of the residual starting material.

### Classical force-field calculations

The formation energy calculations were performed using the large-scale atomic/molecular massively parallel simulator (LAMMPS)[40]. The interaction between copper atoms was described by the classical embedded atom potential (EAM) potential[41]. Periodic condition was applied in the in-plane direction of copper foils, while a vacuum spacing of 20 Å was employed in the out-of-plane direction to avoid periodic imaging interaction. The formation energy of copper foils is defined as

$$E_F = E_T - N\varepsilon_{Cu} \qquad (1)$$

where $E_T$ is the total energies of copper foil. $N$ represents the number of copper atoms in the foil model. $\varepsilon_{Cu}$ is the internal energy of copper atoms in the bulk structure, which can be calculated by

$$\varepsilon_{Cu} = E_{bulk}/N_{Cu} \qquad (2)$$

where $E_{bulk}$ is the energy of the copper bulk and $N_{Cu}$ is the number of copper atoms in the bulk model.

### Characterization

The microstructure of the copper foils was characterized by EBSD using an Oxford NordlysMax2 detector equipped within a JEOL JSM-7800F field-emission SEM (FE-SEM). HRTEM was performed on a JEOL JEM ARM 300CF instrument operating at 300 kV. The morphology of the samples was examined by both optical microscopy (Nikon DS-Fi2), JSM-7800F FE-SEM, and AFM (Dimension Edge, Bruker). STM measurements were carried out on a SPECS NAP-STM system. CD spectra were acquired in diffuse reflectance mode on an Applied Photophysics Chirascan instrument, scanning circular copper foil coupons (10 mm in diameter) over the wavelength range of 250–800 nm. GC-MS analysis was performed using an Agilent 7890 A system. Chiral HPLC analysis was carried out on an Agilent 1200 system equipped with an OJ-H column.

### Data acquisition and processing

The raw EBSD data were processed using Channel 5 software to generate IPF maps for presentation. For graded-orientation analysis, the "Radian" parameter was derived from the step distance between measured points combined with the tube diameter, while the "graded angle" was obtained by calculating the misorientation from the Euler angles at corresponding points. Graphene grain statistics were performed on optical microscopy images acquired at fixed 325 µm intervals. Graphene grains were manually counted and categorized to determine percentages. High-index crystal planes in corresponding regions were indexed with the TexTool software based on EBSD data. Raw Raman spectral data were processed in Wire to extract characteristic peak intensity parameters. Formation energies calculations were performed using the LAMMPS package.

### Reporting summary

Further information on research design is available in the Nature Portfolio Reporting Summary linked to this article.

## Data availability

The data that support the findings of this study are available from Figshare[42] and from the corresponding authors upon request. Unprocessed raw data, including CD, GC-MS, HPLC, STM, SEM, and TEM data,

are provided as Supplementary Data 1. Source data are provided with this paper.

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

## Acknowledgements

This work was supported by the Western China Youth Scholars project (to H.S.), the CAS Interdisciplinary Innovation Team (to H.S.), the National Natural Science Foundation of China (Grant No. 51902306 to D.H. and 52303366 to L.Z.), and Suzhou Laboratory (Grant No. SK-1502-2024-055 to F.D.). L.Z. acknowledges the Teli Fellowship from the Beijing Institute of Technology.

## Author contributions

D.H., H.S. and F.D. conceived and designed the project. H.S. and F.D. supervised the research. L.Z. performed the theoretical analysis of surface orientations and built the atomic structure models of the copper foils. D.H. and Z.L. carried out the annealing experiments and property analysis of the gradient surfaces. Y.D. and D.H. conducted the graphene epitaxy experiments on the gradient substrate. J.D. supervised and performed the asymmetric catalysis experiments. X.L., G.W., and X.H. assisted in the characterization of the copper foils. Y.Z. Conducted the graphene characterization. All authors

participated in result discussions and contributed to the preparation and revision of the manuscript.

## Competing interests

The authors declare no competing interests.
