## [Transparent Peer Review file · Nature Communications]

High-throughput chiral copper foils by curved-surface confinement recrystallization

Corresponding Author: Professor Haofei Shi

Version 0:

Reviewer comments:

Reviewer #1

(Remarks to the Author)

This paper introduces a confinement recrystallization method to create chiral copper foils with continuous surface orientation gradients. The technique involves annealing polycrystalline copper foils within a curved confinement, which drives non-equilibrium abnormal grain growth and eliminates grain boundaries. This study demonstrates that varying the confinement curvature systematically controls the surface orientation evolution, enabling the creation of a diverse library of chiral surfaces in a single process. This approach overcomes the limitations of traditional methods, such as heteroepitaxy or chiral imprinting, which are constrained by low throughput and reliance on templates. The authors showed that specific single-crystal orientations can be replicated from the gradient foils via a seeded growth method. Furthermore, they demonstrated the transfer of chirality from the copper substrate to epitaxially grown graphene, confirming the potential of these materials for applications in asymmetric catalysis and chiral electronics. However, the experimental data are not sufficient to fully support the authors' arguments and there are also several issues to be clarified. Details are listed below.

- 1) The work establishes a qualitative link between curvature and the formation of a chiral surface library. However, a more rigorous, quantitative analysis is needed. The authors should measure the degree of chirality on surfaces produced under different curvatures using a technique like circular dichroism. This would provide a direct, quantitative relationship between the curvature and the resulting chirality, moving the finding from a qualitative observation to a predictable, programmable process.
- 2) While the authors suggested potential applications in asymmetric catalysis and chiral electronics, the only direct application shown is the influence of the chiral copper substrate on the epitaxial graphene grain morphology and orientation. To fully support the authors' claims, the authors should provide experimental evidence in either an asymmetric catalytic reaction or spin-polarized electron transport.
- 3) While the authors convincingly demonstrated the generation of orientation gradients and chiral surfaces through confinement recrystallization, the reproducibility of this approach could be more firmly established. Specifically, performing annealing experiments on multiple copper foils under identical curvature conditions and presenting the statistical distribution of the resulting orientation gradients would strongly emphasize the reproducibility of the method.
- 4) AFM topography would be particularly useful to demonstrate the morphological transformation by directly comparing pre- and post-annealed surfaces (e.g., elimination of grain boundaries and emergence of step-terrace structures). To further substantiate the claim of chiral surface formation, STM measurements are recommended, as they can directly resolve the atomic-scale arrangement and provide unambiguous evidence of surface chirality.

Reviewer #2

(Remarks to the Author)

This work outlines a procedure to prepare chiral Cu surfaces through the confinement annealing of copper foil. By tuning the degree of curvature, the either a gradient of Miller indices or more single crystal behavior can be obtained. Interestingly, when exploring graphene growth on the gradient surface spanning the (8 7 32) to (7 8 32) indices, graphene growth orientation varies between the R and S facets. This strategy for using confinement annealing could be an interesting route to

prepare inherently chiral surfaces for use in applications ranging from heterogeneous catalysis to chiral electronics with the potential to be impactful in the field of chiral substrate preparation.

There are several points throughout the manuscript where there is a connection between high Miller index surfaces and chiral surfaces. This language is imprecise. While it is true that high Miller index surfaces are faceted, it is not merely the presence of facets that induces chirality. Rather, there must be chiral defects, such as kinks, that occur repeatedly across the surface to induce chirality. There needs to be more detailed conversation regarding the crystallographic orientation of the grains within the Cu foil to convey the chirality of the surface. Otherwise, the reader cannot tell from the data as presented if there are truly chiral surface sites or if the surface is merely faceted.

The decision to oxidize the surface post-anneal should be discussed in more detail. It is difficult to discern if the surface is covered in a chiral Cu oxide (see work of McEwen, Gellman, Sykes), which can grow on achiral surfaces, or if the Cu itself is truly chiral. It is critical to explain the role the oxide is playing in the chiral designation of the surface as this could be a factor influencing not only surface structure but surface chirality.

Regarding the chiral graphene portion of the manuscript, the data is insufficient to claim chirality in this system. It is clear that the graphene flakes grow in a different orientation on the surface, but having preferential growth along certain symmetry vectors of the surface does not imply chirality. This would be far more convincing if there was some microscopic data demonstrating how the graphene interacts with polarized light or data that confirms the edge structure. With the data in the manuscript and supplemental information it is hard to confirm that the graphene growth is truly chiral and not just varying orientations of achiral graphene growth.

There are sufficient details to reproduce Cu foil confinement annealing and graphene growth on the prepared surfaces. The data is of a high quality, but is not sufficient to confirm that chiral graphene growth is occurring. As this would be a most impactful finding of the work, this lessens the significance and impact of the current study.

Reviewer #3

(Remarks to the Author)

This is a well-written and innovative study presenting a new curved-surface confinement recrystallization method for producing chiral copper foils with orientation gradients. Haofei Shi et al. successfully demonstrate programmable control of chirality and surface orientation, and further validate chirality transfer through graphene epitaxy. The work is both original and significant, addressing key limitations in scalability and orientation control in the fabrication of chiral metal surfaces. Overall, the proposed method enables high-throughput and large-scale production of chiral metallic surfaces, with major implications for asymmetric catalysis, chiral electronics, and 2D material growth. The extensive characterization using EBSD, TEM, and optical microscopy is thorough and convincingly supports the conclusions. The curvature-dependent analysis is systematic and well-documented. The figures (particularly Figs. 1–4) are visually clear and effectively illustrate the gradual orientation transitions and chirality gradients.

Comments for the authors:

1. The manuscript briefly mentions “non-equilibrium abnormal grain growth” and “formation energy minimization,” yet the underlying mechanisms of chirality formation and propagation under curvature confinement are not fully elucidated. The chirality transition from S to R configurations is qualitatively shown but not quantitatively characterized. The inclusion of quantitative descriptors (e.g., chiral angle, surface asymmetry parameters, or handedness index) would strengthen the analysis and allow for more rigorous comparison between surfaces.
2. While the chirality transfer to graphene is an exciting demonstration, the discussion remains largely descriptive and lacks in-depth analysis. The authors could employ Raman spectroscopy or STM/AFM characterization to confirm the structural and electronic influence of the substrate’s chirality on the graphene layer.

Minor Comments:

- Verify typographical and punctuation errors (e.g., in line 24: “applications However,” should read “applications. However,”).
- In the Methods section, specify cooling rates and annealing durations to enhance reproducibility.
- Include error margins or statistical data in the EBSD and curvature–orientation plots (e.g., Fig. 2c).

This article represents a significant advancement in chiral surface engineering. With deeper mechanistic analysis, quantitative chirality characterization, and expanded discussion of scalability, the study could become a landmark contribution to the field of chiral materials and catalysis.

Version 1:

Reviewer comments:

Reviewer #1

(Remarks to the Author)

The authors have well addressed my concerns with the addition of quantitative chirality metrics, asymmetric catalytic data, and multi-scale microscopic characterization. Therefore, I recommend the publication of this paper.

Reviewer #2

(Remarks to the Author)

This manuscript meticulously details work done to prepare chiral Cu surfaces through confinement annealing of copper foil. Following revisions made in response to reviewer comments, the manuscript clearly details how tuning the degree of curvature, chiral Cu surfaces can be generated. These chiral surfaces could then be used for epitaxially grown graphene, supporting the use of these materials in applications ranging from heterogeneous catalysis to chiral electronics.

The authors carefully responded to all reviewer comments with relevant data and revisions to the manuscript text, strengthening the quantitative analysis of chirality, evidence of reproducibility, and more experimental support for the proposed applications in asymmetric catalysis. The added quantitative chirality characterization, evidence for asymmetric catalysis, statistical data, and revisions to the manuscript text emphasize the novel nature of the work and significance of the results presented herein.

Manuscript ID: NCOMMS-25-65691

Title: High-Throughput Chiral Copper Foils by Confinement Recrystallization

We thank the editor and reviewers for their time and very useful comments in improving the quality of this manuscript. Provided below is our detailed response to each question.

Replied to Reviewers' comments:

Reviewer #1: This paper introduces a confinement recrystallization method to create chiral copper foils with continuous surface orientation gradients. The technique involves annealing polycrystalline copper foils within a curved confinement, which drives non-equilibrium abnormal grain growth and eliminates grain boundaries. This study demonstrates that varying the confinement curvature systematically controls the surface orientation evolution, enabling the creation of a diverse library of chiral surfaces in a single process. This approach overcomes the limitations of traditional methods, such as heteroepitaxy or chiral imprinting, which are constrained by low throughput and reliance on templates. The authors showed that specific single-crystal orientations can be replicated from the gradient foils via a seeded growth method. Furthermore, they demonstrated the transfer of chirality from the copper substrate to epitaxially grown graphene, confirming the potential of these materials for applications in asymmetric catalysis and chiral electronics. However, the experimental data are not sufficient to fully support the authors' arguments and there are also several issues to be clarified. Details are listed below.

Our response:

Thank you very much for your positive and encouraging feedback. We have carefully revised the manuscript in accordance with your suggestions. We believe these revisions thoughtfully address the concerns you raised, and we hope the manuscript now meets the high publication standards of *Nature Communications*.

Q1: The work establishes a qualitative link between curvature and the formation of a chiral surface library. However, a more rigorous, quantitative analysis is needed. The authors should measure the degree of chirality on surfaces produced under different curvatures using a technique like circular dichroism. This would provide a direct, quantitative relationship between the curvature and the resulting chirality, moving the finding from a qualitative observation to a predictable, programmable process.

Our response: Thank you for your valuable comments. Through the proposed curved-surface confinement annealing, grain-boundary-free copper foils with graded surfaces are formed, offering a chiral surface library. In Figure R1, the atomic configurations and chirality degrees of surfaces are displayed in the inverse pole figure. The surfaces denoted by the edges of the stereographic triangle in the inverse pole figure, with the Miller indices of $(m\ m\ n)$, $(m\ n\ n)$ and $(0\ m\ n)$, have a mirror symmetry and therefore are achiral. The deviation of a chiral surface from its nearest achiral counterpart is defined as the degree of chirality, which is dependent on both the magnitude of gradient (out-of-plane orientation) and the direction of gradient (in-plane orientation) in copper surface. During the confinement annealing process, the introduction of curvature primarily affects the magnitude of gradient in copper foil, while the direction of gradient is inherently random. However, it is worth noting that the direction of gradient can be strategically controlled through a well-defined seeded growth, as illustrated in Figure R2. Using a seed with a well-defined orientation, a graded copper foil with a desired degree of chirality can be achieved.

Figure R1. Chirality representation and quantification of copper surfaces. a, Illustration showing the atomic configurations of Cu surfaces. The surfaces denoted by the edges of the stereographic triangle have a mirror symmetry and therefore are achiral. b, the degree of chirality of copper surfaces, which is defined as the deviation of a chiral surface from its nearest achiral counterpart.

Figure R2. Schematics of the controllable synthesis of graded copper surface by seeded growth. a, EBSD map of inverse pole figure. b-c, illustrations showing the formation of graded copper foils with the seed aligned along directions I and II as defined in a.

The chirality of the copper surfaces was assessed using circular dichroism (CD) spectroscopy. As shown in Figure R3, local regions of the curved-annealed copper surface with different Miller indices exhibit distinct CD signals. The Cu(0 27 96)

surface, which is achiral, shows a CD signal close to zero. In contrast, the Cu(60 32 73)^S surface displays a maximum chirality signal of 20 mdeg, while the Cu(14 67 73)^R surface also exhibits a signal with an opposite chirality, yielding a maximum signal near -30 mdeg. These results are fully consistent with the expected chirality for each surface, providing direct experimental evidence linking surface configurations to their chiral properties.

Figure R3. Circular dichroism characterization of gradient copper surfaces. **a**, EBSD IPF maps of the copper surfaces used for Circular dichroism (CD). **b**, Atomic surface models and schematic analysis of the intrinsic chirality for the corresponding crystal planes in (a). **c**, CD measurements spectra measured from the copper foils shown in (a, b).

We have revised Figure 2 and added the following discussion in the revised manuscript:

“The dependence of chirality degree on the surface Miller indices was examined. In Figure 2e-f, the atomic configurations and chirality degrees of surfaces are displayed in the inverse pole figure. The surfaces denoted by the edges of the stereographic triangle in the inverse pole figure, with the Miller indices of (m m n), (m n n) and (0 m n), have a mirror symmetry and therefore are achiral. In contrast, copper surfaces located in the inner part have kinked steps and are truly chiral. The deviation of a chiral surface from its nearest achiral counterpart is defined as the degree of chirality. The greater a chiral surface deviates from its nearest achiral counterpart, the higher its degree of chirality. Figure 2g reveals that the two sides of the graded surface across a mirror-symmetry plane exhibit opposite chirality. It is worth noting that, a graded copper foil with a desired degree of chirality can be achieved by strategically designing

the magnitude of gradient (out-of-plane orientation) and the direction of gradient (in-plane orientation) in copper surface.”

Figure 2. The effect of curvature on surface orientations evolution. a, EBSD IPF maps of copper foils (with arc length of 1.3 mm) annealed under different curvatures (κ). **b,** Schematic diagram of the “graded angle” and the “radian”. **c,** Relationship between graded angle and radian of copper foil with curvature of 0.024 mm⁻¹. Linear fits yield slopes of 1.000 ± 0.003 (R²=1.000). **d,** Relationship between the graded angle and quartz tube diameters, fitted using an allometric model (R² =0.999). **e,** the surfaces denoted by the edges of the stereographic triangle have a mirror symmetry and therefore are achiral. **f,** the degree of chirality of Cu surfaces, which is defined as the deviation of a chiral surface from its nearest achiral counterpart. **g,** Chirality of the graded copper surfaces denoted in inverse pole figure.

We have revised Figure 3 and added the discussion in the revised manuscript

accordingly, as followed:

“The chirality of the curved-annealed copper surfaces was assessed using circular dichroism (CD) spectroscopy. Measurements (Figure 3a, b) reveal distinct CD signals corresponding to different surface configurations. For instance, the achiral Cu(0 27 96) surface shows a near-zero response, whereas chiral surfaces such as Cu(60 32 73)^S and Cu(14 67 73)^R exhibit pronounced signals of opposite chirality (ca. +20 mdeg and -30 mdeg, respectively). The CD response for each surface aligns well with its chirality.”

Figure 3. Characterization of chirality of the copper surfaces and seeded growth of chiral copper foils. a, EBSD IPF maps of the copper surfaces used for Circular dichroism (CD) measurements. b, Atomic surface models and schematic analysis of the intrinsic chirality for the corresponding crystal planes in (a). c, CD spectra measured from the copper foils shown in (a,b). d–g, Photographs and corresponding EBSD inverse pole figure (IPF) maps of single-crystal copper foils fabricated by seeded growth. d, Cu(10 11 13)^R; e, Cu(7 12 14)^R; f, Cu(5 6 18)^R; and g, Cu(11

1 32)^S. Scale bars: 200 μm (d-f), 50 μm (g).

Q2: While the authors suggested potential applications in asymmetric catalysis and chiral electronics, the only direct application shown is the influence of the chiral copper substrate on the epitaxial graphene grain morphology and orientation. To fully support the authors' claims, the authors should provide experimental evidence in either an asymmetric catalytic reaction or spin-polarized electron transport.

Our response: Thank you for your valuable comments. In response to your suggestion, we performed asymmetric catalysis using the gradient copper foils as solid catalysts. The enantioselective oxidation of racemic 1-phenylethanol to acetophenone confirmed that the chiral copper surface induces enantioselectivity, as shown in Figure R4. Chiral HPLC analysis of the recovered 1-phenylethanol revealed an enantiomeric excess (ee) of 5.82% when the chiral Cu(07 69 72) foil was employed. This value is significantly higher than that of the starting material and the control experiment using the achiral Cu(5 6 6) foil. These results provide direct experimental evidence that an intrinsically chiral metallic surface can exhibit enantioselective catalytic activity, demonstrating that curved-surface-annealed copper foils with engineered chirality can serve as solid catalysts to drive enantioselective reactions directly.

We have added relevant discussion in our revised manuscript:

“The asymmetric catalysis experiments demonstrated that curved-surface annealed copper foils can effectively catalyze the oxidation of racemic 1-phenylethanol while inducing enantioselectivity. Gas chromatography-mass spectrometry (GC-MS) spectrum (Supplementary Fig. 14,15) and Chiral High-Performance Liquid Chromatography (HPLC) in Supplementary Fig. 16 confirmed an enantiomeric enrichment (ee = 5.82%) was achieved when the intrinsically chiral Cu(07 69 72) foil was employed as a catalyst. This value is significantly higher than that of the starting material and the control experiment using the achiral Cu(5 6 6) foil. This result provides direct experimental evidence that the chirality-imparted copper substrate exhibits asymmetric catalytic activity.”

Figure R4. Validation of enantioselective catalysis through chiral High-Performance Liquid Chromatography (HPLC) analysis. a, Chiral HPLC chromatogram of the 1-phenylethanol starting material (racemic mixture). **b,** Chiral HPLC chromatogram of recovered 1-phenylethanol from the control reaction using an achiral Cu(5 6 6) as catalyst. **c,** Chiral HPLC chromatogram of recovered 1-phenylethanol from the reaction catalyzed by the chiral Cu(06 69 72).

Q3: While the authors convincingly demonstrated the generation of orientation gradients and chiral surfaces through confinement recrystallization, the reproducibility of this approach could be more firmly established. Specifically, performing annealing experiments on multiple copper foils under identical curvature conditions and

presenting the statistical distribution of the resulting orientation gradients would strongly emphasize the reproducibility of the method.

Our response: Thank you for your constructive suggestion. To directly address the point regarding reproducibility, we have performed three independent repetitions of the annealing experiment under an identical, fixed curvature condition. The results, presented in Figure R5, provide the statistical distribution of the generated orientation gradients across these multiple trials. These data from repeated trials, further supported by experiments on different copper foils (Supplementary Fig. 11), robustly confirms that the confinement annealing approach reliably produces orientation gradients and chiral surfaces. The statistical profile from these repeated experiments matches that of our original findings, firmly establishing the reproducibility of the method.

Figure R5. Reproducibility of annealing on multiple copper foils under curvature ($\kappa = 0.029 \text{ cm}^{-1}$). a-b, Representative EBSD orientation maps. c, Statistical distribution of graded angle with radian. Error bars: ± 1 standard deviation. Linear fits yield slopes of 0.927 ± 0.018 ($R^2=0.998$), 1.094 ± 0.016 ($R^2=0.999$), and 1.000 ± 0.048 ($R^2=0.984$).

Supplementary Fig. 11 Copper foils with a thickness of 46 μm annealed under different curvatures. **a**, Photographs of annealed copper foils, the scale bars are 5 mm. **b-c**, EBSD results, the scale bars are 2 mm in (b). **d**, Relationship between the graded angle with radian. Error bars: ± 1 standard deviation. Linear fits yield slopes of 0.909 ± 0.067 ($R^2=0.979$). **e**, Relationship between the graded angle and quartz tube diameters, fitted using an allometric model ($R^2 = 0.973$).

The relevant content has been updated to the revised manuscript:

“To further verify the reproducibility of the method, Multiple independent experiments were performed under a fixed curvature condition ($\kappa = 0.029 \text{ cm}^{-1}$). The linear relationship between the graded orientation angle and curvature was consistently reproduced across multiple independent trials (Supplementary Fig. 10), providing additional statistical confirmation of the robustness of our confinement annealing approach. Similar quantitative relationships and were observed in copper foils with different thickness of 46 μm (Supplementary Fig. 11), demonstrating the universal applicability of the approach.”

Q4: AFM topography would be particularly useful to demonstrate the morphological transformation by directly comparing pre- and post-annealed surfaces (e.g., elimination of grain boundaries and emergence of step–terrace structures). To further substantiate the claim of chiral surface formation, STM measurements are recommended, as they can directly resolve the atomic-scale arrangement and provide unambiguous evidence

of surface chirality.

Our response: Thank you for these valuable suggestions. AFM characterization was performed to compare the surface morphology before and after curved annealing. As shown in Figure R6, the as-received foil surface exhibits rolling marks and is generally smooth and featureless. After annealing, well-defined step-terrace structures emerge, confirming a clear morphological transformation consistent with grain boundary elimination and surface reconstruction.

Figure R6. Atomic force microscopy (AFM) characterization. a, AFM topography image before annealing. b, AFM images of copper foils after annealing.

To further probe the atomic-scale structure, STM measurements were conducted. Kinked step edges—a structural signature often associated with chirality—are clearly resolved (Figure R7). While direct atomic-resolution imaging of the step-edge configuration was limited under our experimental conditions, the observed kinked morphology provides strong supporting evidence for a structurally distinct surface.

Figure R7. Scanning tunneling microscopy (STM) characterization of chiral copper surfaces. **a-b**, STM topography images of the copper surface, exhibiting atomic-scale chiral reconstructions. The corresponding atomic structural models identifying the R- (**a**) and S-chirality (**b**).

Besides, the chirality is unambiguously confirmed by circular dichroism (CD) spectroscopy (Figure R3), which shows distinct and opposing CD signals for copper surfaces of different chirality.

Figure R8. Circular dichroism characterization of gradient copper surfaces. **a**, EBSD IPF maps of the copper surfaces used for Circular dichroism (CD). **b**, Atomic surface models and schematic analysis of the intrinsic chirality for the corresponding crystal planes in (a). **c**, CD measurements spectra measured from the copper foils shown in (a, b).

Together, the AFM, STM, CD spectrum and Asymmetry catalyze experiment results provide a coherent multiscale evidence chain: from the macroscopic emergence

of step-terrace structures, to the microscale observation of kinked step edges, and finally to the spectroscopic signature of intrinsic chirality, robustly substantiating the formation of chiral copper surfaces.

The relevant content has been updated in our revised manuscript:

“Atomic force microscopy (AFM) characterizations directly compared the surface morphology before and after curved annealing. The as-received copper foil exhibits rolling marks and a smooth, featureless surface. After annealing, well-defined step-terrace structures emerge across the surface (Supplementary Fig. 6). Scanning tunneling microscopy (STM) images further resolve kinked step edges (Supplementary Fig. 7), a structural signature indicative of surface reconstruction and consistent with chiral morphology.”

Reviewer #2 (Remarks to the Author): This work outlines a procedure to prepare chiral Cu surfaces through the confinement annealing of copper foil. By tuning the degree of curvature, the either a gradient of Miller indices or more single crystal behavior can be obtained. Interestingly, when exploring graphene growth on the gradient surface spanning the (8 7 32) to (7 8 32) indices, graphene growth orientation varies between the R and S facets. This strategy for using confinement annealing could be an interesting route to prepare inherently chiral surfaces for use in applications ranging from heterogeneous catalysis to chiral electronics with the potential to be impactful in the field of chiral substrate preparation.

Our response: Thank you for the positive and encouraging evaluation of our work. We are pleased that you find our confinement annealing strategy to be an interesting and potentially impactful route for preparing inherently chiral surfaces.

Q1: There are several points throughout the manuscript where there is a connection between high Miller index surfaces and chiral surfaces. This language is imprecise. While it is true that high Miller index surfaces are faceted, it is not merely the presence of facets that induces chirality. Rather, there must be chiral defects, such as kinks, that occur repeatedly across the surface to induce chirality. There needs to be more detailed conversation regarding the crystallographic orientation of the grains within the Cu foil

to convey the chirality of the surface. Otherwise, the reader cannot tell from the data as presented if there are truly chiral surface sites or if the surface is merely faceted.

Our response: Thank you for your kind suggestion. We totally agree that not all high Miller index surfaces are chiral. As shown in Fig. R1a, the surfaces denoted by the edges of the stereographic triangle, with the Miller indices of $(m\ m\ n)$, $(m\ n\ n)$ and $(0\ m\ n)$, have a mirror symmetry and therefore are achiral. The degree of a chiral surface deviated from its nearest achiral counterpart is defined as the degree of chirality. From Fig. R1b, we see copper surfaces located in the inner part of the inverse pole figure have kinked steps and are chiral, while those located in the edges possess straight steps and are merely faceted.

Figure R1. Chirality representation and quantification of copper surfaces. **a**, Illustration showing the atomic configurations of Cu surfaces. The surfaces denoted by the edges of the stereographic triangle have a mirror symmetry and therefore are achiral. **b**, the degree of chirality of copper surfaces, which is defined as the deviation of a chiral surface from its nearest achiral counterpart.

We have revised the Figure 2 and the added the discussion in revised manuscript:

Figure 2. The effect of curvature on surface orientations evolution. **a**, EBSD IPF maps of copper foils (with arc length of 1.3 mm) annealed under different curvatures (κ). **b**, Schematic diagram of the “graded angle” and the “radian”. **c**, Relationship between graded angle and radian of copper foil with curvature of 0.024 mm⁻¹. Linear fits yield slopes of 1.000 ± 0.003 ($R^2=1.000$). **d**, Relationship between the graded angle and quartz tube diameters, fitted using an allometric model ($R^2 =0.999$). **e**, the surfaces denoted by the edges of the stereographic triangle have a mirror symmetry and therefore are achiral. **f**, the degree of chirality of Cu surfaces, which is defined as the deviation of a chiral surface from its nearest achiral counterpart. **g**, Chirality of the graded copper surfaces denoted in inverse pole figure.

“The dependence of chirality degree on the surface Miller indices was examined. In Figure 2e-f, the atomic configurations and chirality degrees of surfaces are displayed

in the inverse pole figure. The surfaces denoted by the edges of the stereographic triangle in the inverse pole figure, with the Miller indices of $(m\ m\ n)$, $(m\ n\ n)$ and $(0\ m\ n)$, have a mirror symmetry and therefore are achiral. In contrast, copper surfaces located in the inner part have kinked steps and are truly chiral. The deviation of a chiral surface from its nearest achiral counterpart is defined as the degree of chirality. The greater a chiral surface deviates from its nearest achiral counterpart, the higher its degree of chirality. Figure 2g reveals that the two sides of the graded surface across a mirror-symmetry plane exhibit opposite chirality. It is worth noting that, a graded copper foil with a desired degree of chirality can be achieved by strategically designing the magnitude of gradient (out-of-plane orientation) and the direction of gradient (in-plane orientation) in copper surface.”

Q2: The decision to oxidize the surface post-anneal should be discussed in more detail. It is difficult to discern if the surface is covered in a chiral Cu oxide (see work of McEwen, Gellman, Sykes), which can grow on achiral surfaces, or if the Cu itself is truly chiral. It is critical to explain the role the oxide is playing in the chiral designation of the surface as this could be a factor influencing not only surface structure but surface chirality.

Our response: We sincerely thank you for raising this critical point regarding the role of surface oxidation. It allows us to clarify a key aspect of our experimental design and to unequivocally establish that the observed chirality originates from the intrinsic structure of the metallic copper, not from any oxide layer. Regarding the seminal work by McEwen *et al.*^[1] that you mentioned, which elegantly showed how a pre-existing chiral metal surface can template a homochiral oxide with amplified features. Our study addresses a distinct scientific question: we aim to establish the intrinsic chirality of the metallic copper surface itself, created by our confinement annealing process.

The brief oxidation was employed exclusively as a post-processing step to enhance optical contrast for the colorimetric visualization in Figure 1d, since the subtle crystallographic orientation gradients were invisible on pristine copper foil. This oxidation treatment only modified the surface of the copper foil and did not alter its

intrinsic crystal structure. The crystallographic orientation, which defines the chirality, remains unchanged throughout the entire depth of the foil and is not merely a surface phenomenon (Supplementary Fig. 5).

Supplementary Fig. 5 The surface orientation of copper foil (25 μm) across thickness direction.

a, Schematic diagram of surface orientation testing at 5-micrometer intervals across thickness direction. b-d, EBSD IPF maps in normal direction at the position A, B and C marked in (a). The scale bars are 200 μm.

To directly address your central concern, we emphasize that all structural and functional data presented in this work were acquired on pristine, oxide-free copper surfaces. Immediately prior to any key characterization (AFM, STM, EBSD, CD spectroscopy, catalytic testing), the sample underwent in-situ reduction in a hydrogen atmosphere (250 °C, sufficient to remove surface oxide while preserving the metal's step-terrace structure).

To prevent any misunderstanding, we have corrected the relevant description in the manuscript:

“To enhance the optical contrast of the resulting graded chiral surface, the curved copper foil was flattened and heated on a hotplate at 200°C in air for 30-60 seconds to

oxidize the surface. The oxidized surface in Figure 1d displays continuously varying color contrasts, implying the variation of the surface orientations of the copper foil.”

Q3: Regarding the chiral graphene portion of the manuscript, the data is insufficient to claim chirality in this system. It is clear that the graphene flakes grow in a different orientation on the surface, but having preferential growth along certain symmetry vectors of the surface does not imply chirality. This would be far more convincing if there was some microscopic data demonstrating how the graphene interacts with polarized light or data that confirms the edge structure. With the data in the manuscript and supplemental information it is hard to confirm that the graphene growth is truly chiral and not just varying orientations of achiral graphene growth. There are sufficient details to reproduce Cu foil confinement annealing and graphene growth on the prepared surfaces. The data is of a high quality, but is not sufficient to confirm that chiral graphene growth is occurring. As this would be a most impactful finding of the work, this lessens the significance and impact of the current study.

Our response: Thank you for this insightful and critical comment, which helps to clarify a key point in our study. We fully agree that preferential growth along different surface symmetry vectors does not, by itself, constitute proof of chirality.

To directly address this point and provide stronger evidence for chiral edge structure, we have performed angle-resolved polarized Raman spectroscopy on the graphene grains, as now shown in Figure R8. The new data provide two complementary lines of evidence:

1. Linear polarization measurements (Figure R8c–e) reveal a polarization anisotropy ratio $\rho = I(D)_{\min}/I(D)_{\max} = 0.58 \pm 0.13$ when rotating the polarization from 0° to 90° . This value is consistent with a chiral edge structure composed of mixed zigzag and armchair segments, as established in prior literature^[2-5].

Figure R8. Angle-resolved polarized Raman spectroscopy analysis of graphene edges. **a**, SEM image of the measured graphene grain. **b**, 2D band intensity mapping. **c**, Typical Raman spectra from a grain edge area under linear polarization rotated from 0° to 90°. **d**, Corresponding D-band intensity comparison. **e**, D-band intensity mapping under linear polarization. **f**, D-band intensity mapping under left- (LCP) and right-handed circular polarization (RCP). **g**, Typical Raman spectra under LCP and RCP. **h**, Corresponding D-band intensity comparison.

2. Circular polarization measurements (Figure R8f–h) show a clear difference in D-band intensity under left-handed (LCP) versus right-handed (RCP) circularly polarized light. This distinct optical response under opposite helicities provides direct spectroscopic evidence of structural chirality at the graphene edge.

Together, these polarized Raman results go beyond simple orientation mapping and demonstrate that the graphene edges exhibit structural chirality, which correlates with the underlying chiral Cu surface.

We have revised the Figure 4 and added following discussion in the revised manuscript:

Figure 4. Epitaxial graphene on the obtained gradient copper foil. **a**, The atomic structure of the chiral surface for graphene epitaxy, with index evolving from $\text{Cu}(8\ 7\ 32)^{\text{S}}$ to $\text{Cu}(5\ 3\ 21)^{\text{S}}$, $\text{Cu}(7\ 2\ 27)^{\text{S}}$, $\text{Cu}(1\ 0\ 4)$, $\text{Cu}(2\ 7\ 27)^{\text{R}}$, $\text{Cu}(3\ 5\ 21)^{\text{R}}$, and $\text{Cu}(7\ 8\ 32)^{\text{R}}$. The scale bars are $10\ \mu\text{m}$. **b**, The typical SEM images of graphene grains. **c**, Statistical results of the occurrence proportion of Graphene Grains A and B along the gradient substrate surface, curves for percentage of grains A and B were fitted by Langevin model with R^2 of 0.991. **d**, SEM images of the graphene grain for Angle-Resolved Raman testing. **e**, 2D band Raman mapping of the graphene grain. **f**, Corresponding D-band intensity comparison under $0\text{-}90^\circ$ polarized light and **g**, under left-handed (LCP) and right-handed circular polarization (RCP) light.

“Angle-resolved polarized Raman spectroscopy on the graphene grains reveals that the graphene grain edges comprise a hybrid of zigzag and armchair configurations. The Scanning Electron Microscopy (SEM) images and Raman mapping of the graphene grain without polarization shows in Figure 4d,e and Supplementary Fig. 13a. Under linearly polarized light (Figure 4f and Supplementary Fig. 13b,c), the polarization-

dependent from D-band intensity showed a polarization anisotropy ratio ρ ($I(D)_{\min}/I(D)_{\max}$) around 0.58 ± 0.13 under 0-90° polarized light, indicating a chirality edge structure composited by zigzag and armchair^[36-39]. Furthermore, a distinct intensity variation was observed under left-handed (LCP) and right-handed circular polarization (RCP) as shown in Figure 4g and Supplementary Fig. 13d,e, providing complementary evidence for the chiral and hybrid nature of the graphene edges.”

Reviewer #3 (Remarks to the Author): This is a well-written and innovative study presenting a new curved-surface confinement recrystallization method for producing chiral copper foils with orientation gradients. Haofei Shi et al. successfully demonstrate programmable control of chirality and surface orientation, and further validate chirality transfer through graphene epitaxy. The work is both original and significant, addressing key limitations in scalability and orientation control in the fabrication of chiral metal surfaces. Overall, the proposed method enables high-throughput and large-scale production of chiral metallic surfaces, with major implications for asymmetric catalysis, chiral electronics, and 2D material growth. The extensive characterization using EBSD, TEM, and optical microscopy is thorough and convincingly supports the conclusions. The curvature-dependent analysis is systematic and well-documented. The figures (particularly Figs. 1-4) are visually clear and effectively illustrate the gradual orientation transitions and chirality gradients. Comments for the authors:

Our response: Thank you for the highly positive and supportive evaluation of our work. We are delighted that you consider our study innovative and significant and acknowledges the programmable control and high-throughput fabrication potential of the proposed method. In response to your comments—along with those from the other reviewers—we have comprehensively revised the manuscript to enhance the clarity and rigor of the presentation. We deeply appreciate your encouragement and support.

Q1: The manuscript briefly mentions “non-equilibrium abnormal grain growth” and “formation energy minimization,” yet the underlying mechanisms of chirality formation and propagation under curvature confinement are not fully elucidated.

Our response: Thank you for your valuable comments. The structural evolution of

copper foil during annealing is driven by minimization of surface energy, grain boundary energy and elastic energy. In our proposed confinement setup, the thin foil easily adopted the arc-shape of the tube, thereby storing elastic energy. During the following annealing process, this elastic energy was released, and structure was governed by the competition between surface energy and grain boundary energy. For the fully-relaxed arc-shape copper foil, there are two possible structures: (i) a polycrystalline in which multiple grains stitching together by grain boundaries to accommodate the arc shape; (ii) a single-crystal foil with a graded surface, whose normal direction rotates gradually around the arc. Figure R9 illustrated several copper polycrystals with different twin boundaries and low-index surface. Despite twin grain boundaries generally have lower energies than conventional high-angle grain boundaries^[6], and low-index surfaces possess lower surface energies than high-index surfaces^[7], these twined polycrystals still show higher energies than single-crystal graded copper foils (Figure R9d). Therefore, driven by the minimization of formation energy, the single-crystal region ultimately evolved into a large single crystal via abnormal grain growth, exhibiting a graded surface.

We have added the following discussion in the revised manuscript:

“The structural evolution of copper foil was driven by the minimization of surface energies, grain boundary energies, and elastic energies. During our proposed confinement annealing process, the copper foil adopted to the arc shape of the tube, storing elastic energy that was later released. For the fully-relaxed structure, two stable configurations were possible: (i) a polycrystal with grain boundaries accommodating the curvature, or (ii) a single crystal with a gradually rotating surface normal. Despite the graded surface of single-crystal copper foil contains high-energy surface regions, its overall formation energy remains lower than that of polycrystals. Consequently, abnormal grain growth was promoted within the single-crystal region, leading to the formation of a large-area single-crystal copper foil with a graded surface (Supplementary Fig. 2).”

Figure R9. Formation energies of arc-shape copper foils with polycrystalline or single-crystal structures. **a-b**, Atomic structures of arc-shape polycrystals with twin boundaries and low-index surfaces **c**, Atomic structures of arc-shape single crystals with a graded surface under different curvatures. **d**, Formation energies of different arc-shape copper foils shown in a-c.

Q2: The chirality transition from S to R configurations is qualitatively shown but not quantitatively characterized. The inclusion of quantitative descriptors (e.g., chiral angle, surface asymmetry parameters, or handedness index) would strengthen the analysis and allow for more rigorous comparison between surfaces.

Our response: Thank you for your valuable comments. As shown in Fig. R1a, the surfaces denoted by the edges of the stereographic triangle, with the Miller indices of $(m\ m\ n)$, $(m\ n\ n)$ and $(0\ m\ n)$, have a mirror symmetry and therefore are achiral. Other surfaces located in the inner part of the inverse pole figure are chiral, where the handedness is labeled. Besides, the deviation of a chiral surface from its nearest achiral counterpart is defined as the degree of chirality. From Fig. R1b, we see the greater a chiral surface deviates from its nearest achiral counterpart, the higher its degree of

chirality.

Figure R1. Chirality representation and quantification of copper surfaces. a, Illustration showing the atomic configurations of Cu surfaces. The surfaces denoted by the edges of the stereographic triangle have a mirror symmetry and therefore are achiral. b, the degree of chirality of copper surfaces, which is defined as the deviation of a chiral surface from its nearest achiral counterpart.

We have revised the Figure 2 and added the following discussion in the revised manuscript:

Figure 2. The effect of curvature on surface orientations evolution. **a**, EBSD IPF maps of copper foils (with arc length of 1.3 mm) annealed under different curvatures (κ). **b**, Schematic diagram of the “graded angle” and the “radian”. **c**, Relationship between graded angle and radian of copper foil with curvature of 0.024 mm⁻¹. Linear fits yield slopes of 1.000 ± 0.003 ($R^2=1.000$). **d**, Relationship between the graded angle and quartz tube diameters, fitted using an allometric model ($R^2 = 0.999$). **e**, the surfaces denoted by the edges of the stereographic triangle have a mirror symmetry and therefore are achiral. **f**, the degree of chirality of Cu surfaces, which is defined as the deviation of a chiral surface from its nearest achiral counterpart. **g**, Chirality of the graded copper surfaces denoted in inverse pole figure.

“The dependence of chirality degree on the surface Miller indices was examined. In Figure 2e-f, the atomic configurations and chirality degrees of surfaces are displayed in the inverse pole figure. The surfaces denoted by the edges of the stereographic triangle in the inverse pole figure, with the Miller indices of (m m n), (m n n) and (0 m

n), have a mirror symmetry and therefore are achiral. In contrast, copper surfaces located in the inner part have kinked steps and are truly chiral. The deviation of a chiral surface from its nearest achiral counterpart is defined as the degree of chirality. The greater a chiral surface deviates from its nearest achiral counterpart, the higher its degree of chirality. Figure 2g reveals that the two sides of the graded surface across a mirror-symmetry plane exhibit opposite chirality. It is worth noting that, a graded copper foil with a desired degree of chirality can be achieved by strategically designing the magnitude of gradient (out-of-plane orientation) and the direction of gradient (in-plane orientation) in copper surface.”

Q3: While the chirality transfer to graphene is an exciting demonstration, the discussion remains largely descriptive and lacks in-depth analysis. The authors could employ Raman spectroscopy or STM/AFM characterization to confirm the structural and electronic influence of the substrate’s chirality on the graphene layer.

Our response: Thank you for raising this important point regarding the need for deeper mechanistic analysis of chirality transfer. In direct response to the suggestion to employ Raman spectroscopy and STM/AFM characterization, we have performed a series of complementary experiments. These studies structurally link the chirality of the copper substrate to that of the epitaxial graphene, moving beyond a descriptive discussion to provide direct evidence for the transfer mechanism.

1. Atomic-Scale Chiral Template of the Copper Substrate

To establish the structural origin of the chirality transfer, we first solidified the evidence for an atomically chiral copper surface. AFM images in Figure R10a,b confirms that the curved annealing process transforms the initially smooth, rolled foil surface into one with distinct step-terrace structures. Crucially, STM images in Figure R10c,d resolves these features at the atomic scale, revealing serrated (kinked) step edges with clear structural asymmetry. These features are direct topographic signatures of a chiral metal surface and provide the essential atomic-scale chiral template for subsequent graphene epitaxy.

Figure R10. AFM and STM characterization of chiral copper surfaces. a-b, AFM topography images before annealing (**a**) and after annealing (**b**). **c-d,** STM topography images of the copper surface, exhibiting atomic-scale chiral reconstructions. The corresponding atomic structural models identifying the R- (**c**) and S-chirality (**d**).

2. Probing Chirality in Epitaxial Graphene Edges

To investigate how this chiral template influences the overlayer, we characterized the synthesized graphene grains. While AFM (Figure R11)) reveals the grain morphology, its resolution is insufficient for definitive atomic-edge analysis. Therefore, we employed angle-resolved polarized Raman spectroscopy (Figure R8) as a sensitive probe of edge structure and chirality.

Figure R11. AFM image of graphene grains grown on the chiral copper substrate.

Figure R8. Angle-resolved polarized Raman spectroscopy analysis of graphene edges. a, SEM image of the measured graphene grain. b, 2D band intensity mapping. c, Typical Raman spectra from a grain edge area under linear polarization rotated from 0° to 90° . d, Corresponding D-band intensity comparison. e, D-band intensity mapping under linear polarization. f, D-band intensity mapping under left- (LCP) and right-handed circular polarization (RCP). g, Typical Raman spectra under LCP and RCP. h, Corresponding D-band intensity comparison.

The Raman analysis confirms chiral edge structures in the graphene. Linear Polarization (Figure R8c-e) imply that the D-band intensity exhibits significant

anisotropy. The anisotropy ratio, defined as $\rho = I(D)_{\min}/I(D)_{\max}$, is measured to be approximately 0.58 ± 0.13 . This value is consistent with edges composed of a mixture of zigzag and armchair segments, indicative of a chiral edge structure^[2-4]. Circular Polarization (Figure R8f-h) displays a pronounced difference in the D-band intensity under left-handed (LCP) versus right-handed circularly polarized (RCP) light. This provides direct spectroscopic evidence for structural chirality at the graphene edges.

The key link between the substrate and graphene chirality is established through the epitaxial growth mechanism. It is well-known that graphene edges nucleate and align along the atomic steps of the copper substrate^[8,9]. Our STM data directly show that the curved-annealed copper possesses kinked, chiral step edges. Concurrently, our polarized Raman data demonstrate that the resulting graphene edges exhibit a mixed edge structure with chiral character.

We have revised Figure 4 and integrated this strengthened analysis into the revised manuscript:

“Atomic force microscopy (AFM) characterizations directly compared the surface morphology before and after curved annealing. The as-received copper foil exhibits rolling marks and a smooth, featureless surface. After annealing, well-defined step-terrace structures emerge across the surface (Supplementary Fig. 6). Scanning tunneling microscopy (STM) images further resolve kinked step edges (Supplementary Fig. 7), a structural signature indicative of surface reconstruction and consistent with chiral morphology.”

“Angle-resolved polarized Raman spectroscopy on the graphene grains reveals that the graphene grain edges comprise a hybrid of zigzag and armchair configurations. The Scanning Electron Microscopy (SEM) images and Raman mapping of the graphene grain without polarization shows in Figure 4d,e and Supplementary Fig. 13a. Under linearly polarized light (Figure 4f and Supplementary Fig. 13b,c), the polarization-dependent from D-band intensity showed a polarization anisotropy ratio ρ ($I(D)_{\min}/I(D)_{\max}$) around 0.58 ± 0.13 under 0-90° polarized light, indicating a chirality edge structure composited by zigzag and armchair^[36-39]. Furthermore, a

distinct intensity variation was observed under left-handed (LCP) and right-handed circular polarization (RCP) as shown in Figure 4g and Supplementary Fig. 13d,e, providing complementary evidence for the chiral and hybrid nature of the graphene edges.”

Figure 4. Epitaxial graphene on the obtained gradient copper foil. a, The atomic structure of the chiral surface for graphene epitaxy, with index evolving from $\text{Cu}(8\ 7\ 32)^{\text{S}}$ to $\text{Cu}(5\ 3\ 21)^{\text{S}}$, $\text{Cu}(7\ 2\ 27)^{\text{S}}$, $\text{Cu}(1\ 0\ 4)$, $\text{Cu}(2\ 7\ 27)^{\text{R}}$, $\text{Cu}(3\ 5\ 21)^{\text{R}}$, and $\text{Cu}(7\ 8\ 32)^{\text{R}}$. The scale bars are $10\ \mu\text{m}$. **b,** The typical SEM images of graphene grains. **c,** Statistical results of the occurrence proportion of Graphene Grains A and B along the gradient substrate surface, curves for percentage of grains A and B were fitted by Langevin model with R^2 of 0.991. **d,** SEM images of the graphene grain for Angle-Resolved Raman testing. **e,** 2D band Raman mapping of the graphene grain. **f,** Corresponding D-band intensity comparison under $0\text{-}90^\circ$ polarized light and **g,** under left-handed (LCP) and right-handed circular polarization (RCP) light.

Minor Comments:

Q1: Verify typographical and punctuation errors (e.g., in line 24: “applications However,” should read “applications. However,”).

Our response: Thank you for the careful review of our manuscript. All typographical and punctuation errors have been carefully identified and corrected throughout the text. We sincerely appreciate your suggestions, which have helped improve the quality of the manuscript.

Q2: In the Methods section, specify cooling rates and annealing durations to enhance reproducibility.

Our response: We appreciate your kind suggestion. In response, we have added specific details regarding the cooling rates and annealing durations in the Methods section to ensure the reproducibility of our approach. These clarifications are now incorporated in the revised manuscript:

“At 1060°C, the system was held under a gas flow of 100 sccm H₂ and 500 sccm Ar for 30 minutes. After ceasing the H₂ flow, the furnace was opened, and the sample was then cooling down to room temperature within 30 minutes (~35°C/min) under 500 sccm Ar.”

Q3: Include error margins or statistical data in the EBSD and curvature–orientation plots (e.g., Fig. 2c).

Our response: Thank you for this important suggestion. In response, we have supplemented error margins (representing standard deviation) and key statistical data for the quantitative plots in the caption of each Figure, includes the specific figures mentioned (Figure 2c,d, Figure 3c, Figure 4c, Supplementary Fig. 9c,b, Supplementary Fig. 10c, Supplementary Fig. 11d,e). The details of fitting parameters (e.g., slope, fitted model), as well as goodness-of-fit metrics (e.g., R^2) are listed in the Source Data file.

Comments: This article represents a significant advancement in chiral surface engineering. With deeper mechanistic analysis, quantitative chirality characterization, and expanded discussion of scalability, the study could become a landmark contribution to the field of chiral materials and catalysis.

Our Response: We are deeply grateful for your highly encouraging and constructive assessment of our work. We sincerely appreciate the recognition that our study could represent a significant advance in the field. In direct response to the valuable suggestions for improvement, we have substantially strengthened the manuscript by: (i) providing a deeper mechanistic analysis of the curvature-driven orientation gradient formation; (ii) incorporating quantitative chirality characterization (e.g., via circular dichroism spectroscopy and enantiomeric excess calculations in catalysis); and (iii) offering more profound mechanistic insights into the chirality transfer and evolution during the process. We believe these revisions have significantly enhanced the impact and clarity of our study. Thank you again for guiding these crucial improvements.

Replied to Editor's Comments:

Editor's Comments: In particular, the reviewers perceived the proposed concept for high-throughput confinement recrystallization of copper foils as not conclusive enough. Reviewer #1 finds the work significant but emphasizes that the current experimental data are insufficient to substantiate the authors' claims. They highlight the need for quantitative analysis of chirality, stronger experimental support for proposed applications, and clearer evidence of reproducibility through statistical validation. Reviewer #2 questions the accuracy of the mechanistic descriptions, particularly the connection between surface faceting and chirality, and calls for clarification on the role of surface oxidation, as well as additional microscopic and optical data to verify true surface and graphene chirality. Reviewer #3 appreciates the study's significance but notes a lack of in-depth analysis, recommending complementary techniques, quantitative chirality characterization, and expanded methodological and statistical details to strengthen the reproducibility. Please let us clarify that our interest in a method for chiral metallic foils as well as deeper mechanistic insight into the control of surface chirality persists, but this has to be substantiated by further unequivocal experimental evidence and mechanistic descriptions.

Our response: We sincerely thank the Editor and the Reviewers for the constructive

and insightful comments, which have significantly helped us to improve the quality and clarity of our manuscript. In response to all concerns raised, we have conducted further experiments, analyses, and textual revisions. All changes have been highlighted in the revised manuscript.

The reviewers suggested steps to address their concerns:

* Provide more quantitative experimental evidence of surface chirality; for instance, include circular dichroism or equivalent measurements, repeated annealing experiments, and statistical validation (#1, #3).

Our response: We established a quantitative “degree of chirality” metric based on surface crystallographic (Figure 2 in the revised manuscript). Circular dichroism (CD) spectroscopy directly revealed clear optical activity from the intrinsic chiral structures of our annealed foils (Figure 3 in the revised manuscript). The robustness and reproducibility of our method were statistically validated through multiple independent replications of the entire annealing process under identical conditions (Supplementary Fig. 10 in the revised manuscript).

* Clarify the mechanistic basis of chirality: Distinguish faceting from true chiral features, explain the role of oxidation, and add microscopic or optical data to confirm genuine chiral behavior in both Cu surfaces and graphene (#1, #2).

Our response: Both chiral and achiral surfaces have been clarified in the revised manuscript. Copper surfaces located in the inner part of the inverse pole figure have kinked steps and are chiral, while those located in the edges possess straight steps and are merely faceted (Figure 2 in the revised manuscript). We cleared that brief oxidation step exclusively for optical visualization (in line 81-86 in the revised manuscript). The intrinsic chiral activity of copper surface was demonstrated by CD spectroscopy and enantioselective catalysis (Figure 3 and Supplementary Fig. 14-16 in the revised manuscript), while that of graphene was confirmed by Angle-resolved polarized Raman spectroscopy (Figure 4 in the revised manuscript).

* Strengthen structural and mechanistic analysis by incorporating complementary techniques such as AFM, STM, and Raman spectroscopy to validate morphology, atomic structure, and the influence of chirality on graphene growth (#1, #3).

Our response: To systematically address this concern, we employed a suite of complementary techniques, establishing a coherent evidence chain from morphology and atomic structure to intrinsic chirality and its transfer. AFM confirmed the step-terrace configuration of annealed copper surface (Supplementary Fig. 6 in the revised manuscript), and STM provided microscale evidence that the steps are generally kinked (Supplementary Fig. 7), suggesting the chirality of copper surface. Angle-resolved polarized Raman spectroscopy performed on the grown graphene revealed distinct polarization anisotropy and a differential response to circularly polarized light in the D band (Figure 4 in the revised manuscript). This provides direct spectroscopic evidence for chiral graphene, indicating the chirality duplication from underlying copper template.

References

1. Schilling, A. C., Therrien, A. J., Hannagan, R. T. et al., Templated Growth of a Homochiral Thin Film Oxide, *Acs Nano* 14,4682-4688 (2020).
2. Tao, C. G., Jiao, L. Y., Yazyev, O. V. et al., Spatially resolving edge states of chiral graphene nanoribbons, *Nature Physics* 7,616-620 (2011).
3. You, Y. M., Ni, Z. H., Yu, T. et al., Edge chirality determination of graphene by Raman spectroscopy, *Applied Physics Letters* 93,(2008).
4. Casiraghi, C., Hartschuh, A., Qian, H. et al., Raman Spectroscopy of Graphene Edges, *Nano Letters* 9,1433-1441 (2009).
5. Gorbachev, R. V., Song, J. C. W., Yu, G. L. et al., Detecting topological currents in graphene superlattices, *Science* 346,448-451 (2014).
6. Murr, L. E., *Interfacial phenomena in metals and alloys*, (1975).
7. Jin, S., Huang, M., Kwon, Y. et al., Colossal grain growth yields single-crystal metal foils by contact-free annealing, *Science* 362,1021-1025 (2018).
8. Gao, J. F., Yip, J., Zhao, J. J. et al., Graphene Nucleation on Transition Metal Surface: Structure Transformation and Role of the Metal Step Edge, *Journal of the American Chemical Society* 133,5009-5015 (2011).
9. Wu, R. Z., Ding, Y., Yu, K. M. et al., Edge-Epitaxial Growth of Graphene on Cu

with a Hydrogen-Free Approach, *Chemistry of Materials* 31,2555-2562 (2019).

Manuscript ID: NCOMMS-25-65691A

Title: High-Throughput Chiral Copper Foils by Confinement Recrystallization

We thank the editor and reviewers for their time and very useful comments in improving the quality of this manuscript. Provided below is our detailed response to each question.

Replied to Reviewers' comments:

Reviewer #1 (Remarks to the Author):

The authors have well addressed my concerns with the addition of quantitative chirality metrics, asymmetric catalytic data, and multi-scale microscopic characterization. Therefore, I recommend the publication of this paper.

Our response:

Thank you very much for your careful review of our revised manuscript and for your positive feedback. We are delighted that you found our additional quantitative chirality metrics, asymmetric catalysis data, and multi-scale microscopic characterization to have adequately addressed your previous concerns. Your valuable suggestions were instrumental in enhancing the depth and rigor of our study, making the characterization of the chiral surfaces and the demonstration of their applications more solid. We sincerely appreciate your insightful comments and effort you dedicated to this process.

Thank you again for your recognition and support of our work, and for recommending its publication.

Reviewer #2 (Remarks to the Author):

This manuscript meticulously details work done to prepare chiral Cu surfaces through confinement annealing of copper foil. Following revisions made in response to reviewer comments, the manuscript clearly details how tuning the degree of curvature, chiral Cu surfaces can be generated. These chiral surfaces could then be used for epitaxially grown graphene, supporting the use of these materials in

applications ranging from heterogeneous catalysis to chiral electronics.

The authors carefully responded to all reviewer comments with relevant data and revisions to the manuscript text, strengthening the quantitative analysis of chirality, evidence of reproducibility, and more experimental support for the proposed applications in asymmetric catalysis. The added quantitative chirality characterization, evidence for asymmetric catalysis, statistical data, and revisions to the manuscript text emphasize the novel nature of the work and significance of the results presented herein.

Our response:

We sincerely thank you for your thorough review of the revised version of our manuscript and for your positive assessment. We greatly appreciate your acknowledgment of our work on successfully preparing chiral Cu surfaces through systematic curvature tuning and its potential for applications such as heterogeneous catalysis and chiral electronics. Your comments were crucial in guiding us strengthened the manuscript by supplementing quantitative chirality analysis, evidence for asymmetric catalysis, statistical data, and corresponding textual revisions, thereby significantly improving the completeness and persuasiveness of the study.

We are deeply grateful for your professional guidance and constructive suggestions throughout the review process, which played a key role in enhancing the quality and presentation of this research. Your affirmation and support are a great encouragement to us.

Thank you once again for your valuable contribution.